# DIC-ST: A Hybrid Prediction Framework Based on Causal Structure Learning for Cellular Traffic and Its Application in Urban Computing

**Kaisa Zhang** , **Gang Chuai \***, **Jinxi Zhang** , **Xiangyu Chen** , **Zhiwei Si and Saidiwaerdi Maimaiti**

Department of Information and Communication Engineering, Beijing University of Posts and Telecommunications, Beijing 100876, China; kaisa@bupt.edu.cn (K.Z.); jinxi@bupt.edu.cn (J.Z.); xychen324@bupt.edu.cn (X.C.); sizhiwei@bupt.edu.cn (Z.S.); saidi216@bupt.edu.cn (S.M.)
\* Correspondence: chuai@bupt.edu.cn

**Abstract:** The development of technology has strongly affected regional urbanization. With development of mobile communication technology, intelligent devices have become increasingly widely used in people's lives. The application of big data in urban computing is multidimensional; it has been involved in different fields, such as urban planning, network optimization, intelligent transportation, energy consumption and so on. Data analysis becomes particularly important for wireless networks. In this paper, a method for analyzing cellular traffic data was proposed. Firstly, a method to extract trend components, periodic components and essential components from complex traffic time series was proposed. Secondly, we introduced causality data mining. Different from traditional time series causality analysis, the depth of causal mining was increased. We conducted causality verification on different components of time series and the results showed that the causal relationship between base stations is different in trend component, periodic component and essential component in urban wireless network. This is crucial for urban planning and network management. Thirdly, DIC-ST: a spatial temporal time series prediction based on decomposition and integration system with causal structure learning was proposed by combining GCN. Final results showed that the proposed method significantly improves the accuracy of cellular traffic prediction. At the same time, this method can play a crucial role for urban computing in network management, intelligent transportation, base station siting and energy consumption when combined with remote sensing map information.

**Keywords:** urban computing; cellular traffic; EMD; causal structure learning; GCN; smart city

## 1. Introduction

With the development of urbanization, a large number of people gather in the city, while cities provide people with an environment for work, study and life; additionally, problems such as traffic congestion and environmental degradation appear. Traditional urban management schemes can no longer cope with this situation. In this context, the concept of smart city came into being. Urban computing is a core technology in smart city construction. Urban computing is a new cross field of urban planning, transportation, energy, environment, economy and sociology based on computer science. The task of urban computing is to first perceive and obtain all kinds of big data generated in the city, and then analyze, process and display big data by using efficient data management technology, advanced algorithms and novel visualization method, so as to solve many problems and challenges existing in the city, such as traffic congestion, poor network quality, backward planning and so on. Urban computing based on mobile communication data is an important means to promote urban intelligence. Compared with other data sources, the advantage of wireless network data is to increase non-public transport users information, such as private cars, bicycles in passenger flow analysis or real-time traffic analysis. Urban planning can also be carried out in a more accurate and comprehensive

way. Most importantly, with the increasing number of mobile users, mobile communication has become an indispensable part of people's life. The accurate analysis of wireless network data is the basis for providing users with high-quality multimedia services, so as to ensure the requirements of users for high precision and low delay in different application scenarios such as automatic driving, telemedicine, AR, VR, etc. Urban computing is an emerging discipline to improve the quality of people's life in the city [1].

Wireless network data is divided into signaling data and billing data. Signaling data can be divided into call detail record (CDR) and signal data, including user ID, base station ID, longitude and latitude, etc. The billing data records the user's service start and end time, traffic volume and other information. The large amount of mobile data and long observation cycle can be used to analyze the people footprint in cities on an unprecedented scale [2,3]. In [4], authors described the research progress of wireless traffic in crowd mobility, geographical zoning, urban planning, development and security and privacy. In this paper, we use real wireless network data from an operator in China, and the data was collected by 22 base stations in an area. Figure 1 showed the area where 22 base stations are located. This data collects the traffic information of each base station in the region for 100 days. The joint analysis of cellular traffic prediction and remote sensing map as shown in Figure 1 can help us accurately sort out mobility of the city. The combination of wireless network data and remote sensing data has many advantages; for traffic prediction, because wireless network data is often the statistical value of regional data, these data are not based on the road as vehicular traffic flow data. Combined with remote sensing map, traffic congestion can be quickly located according to wireless network data. For wireless network optimization, traditional network management requires manual positioning of areas with poor network quality. Using remote sensing data for network management can accurately optimize the network in a visual way. Wireless network data often appear in big data analysis technology in a form of time series. Time series prediction is one of the important research topics in the field of data analysis [5]. Time series prediction research involves various fields, such as economy, society, energy, environment, climate and other research fields [6–8].

There are three methods for time series analysis and prediction: traditional statistical models, artificial intelligence models and hybrid models. Due to the complexity and randomness of cellular network data, wireless network traffic data can not be well analyzed by the traditional model. However, although the traffic data in wireless network is complex, it has its own unique characteristics, such as tidal, seasonality, etc. [9]. We proposed a method to extract trend component, periodic component and essential component from complex time series. Then, according to the idea of hybrid model, the decomposition and integration the prediction scheme is adopted to predict three different components, respectively, and finally three prediction results are combined into the final prediction result. In the prediction of three components, we introduced causal data mining. Different from the traditional temporal causal analysis, we increased the depth of causal mining, not for causal verification of time series, but for different components of same time series. Final results showed that in urban wireless networks, the causality correlation between base stations is different in the trend component, periodic component and essential component. This verification result is important for network management and construction. This deep data mining can not only enhance the accuracy of prediction, but also play a role in urban planing, citywide network construction and resource planning.

Causal structure learning was introduced to find the relationship between different time series, and the purpose is to use the relationship between different time series for prediction performance enhancement. The effectiveness of causality for time series prediction has been confirmed in some studies [10,11]. In this paper, we proposed deep causal mining for different components in time series. In the research of smart city, big data temporal spatial analysis is an important part. We decomposed time series data into three components. Each component plays a different role in urban computing. The trend component represents the overall number of users in the region. The research on the trend component can

be used in base station construction, urban planning and so on. The periodic component symbolizes the law of population flow and can be used for wireless network management and traffic information monitoring. The essential component is the key to ensure the quality of user service. We establish a temporal–spatial model for each component and introduce causal structure learning into the model. For different components, the correlation between regions in the city can be found, which can be used for data prediction and analysis in urban computing.

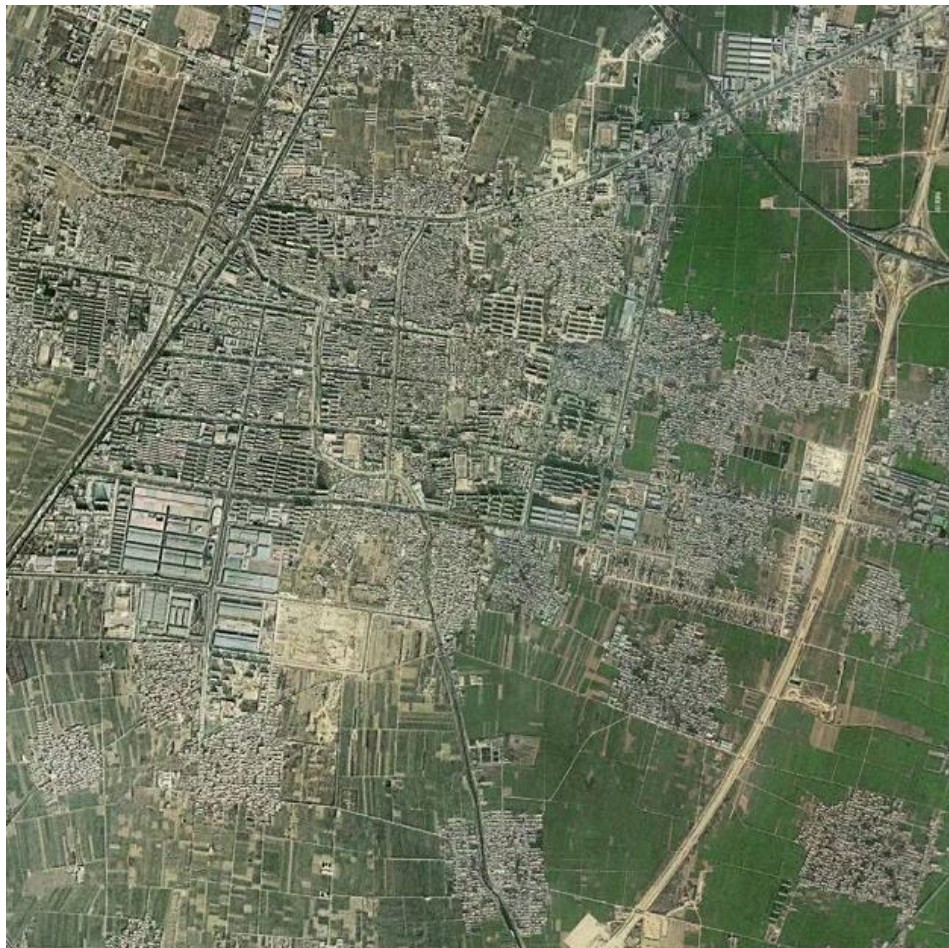

**Figure 1.** Remote sensing map of data collection area.

As mentioned earlier, decomposition integration system and causal mining are combined in this paper. System structure was shown in Figure 2. Traffic data of each base station in the region is decomposed into multiple subsequences. The first integration step is to clustering three components of data. Causal structure learning was used to deal with each component of different regions separately. After causal structure for different components was obtained, this structure can be used for time series prediction. The second integration part is to combine the three prediction results into the final prediction result. A spatial–temporal time series prediction based on decomposition and integration system with causal structure learning (DIC-ST) was proposed. Actual data was used to verify the algorithm and compare with other commonly used algorithms. The main contributions of this paper are as follows:

1. Based on empirical mode decomposition (EMD), a time series clustering method based on spectrum information and information entropy was proposed. This method can extract the cellular network traffic data into three components: periodic component, trend component and essential component.

2. We propose a deep causal mining method for time series data. The traditional time series causal analysis was directly applied to multiple time series, ignoring the different components of time series. In our research, different components contained in time series are used for causal structure learning, respectively. The final results show that the causal structure of each component is different. This deep causal mining helps to clearly sort out the traffic relationship between regions and improve the prediction performance.

3. In order to make wireless network data better used in urban computing research. For cellular traffic data, we proposed a novel time series analysis method DIC-ST. This method improved the accuracy of prediction, and it can serve the construction of smart city from many aspects.

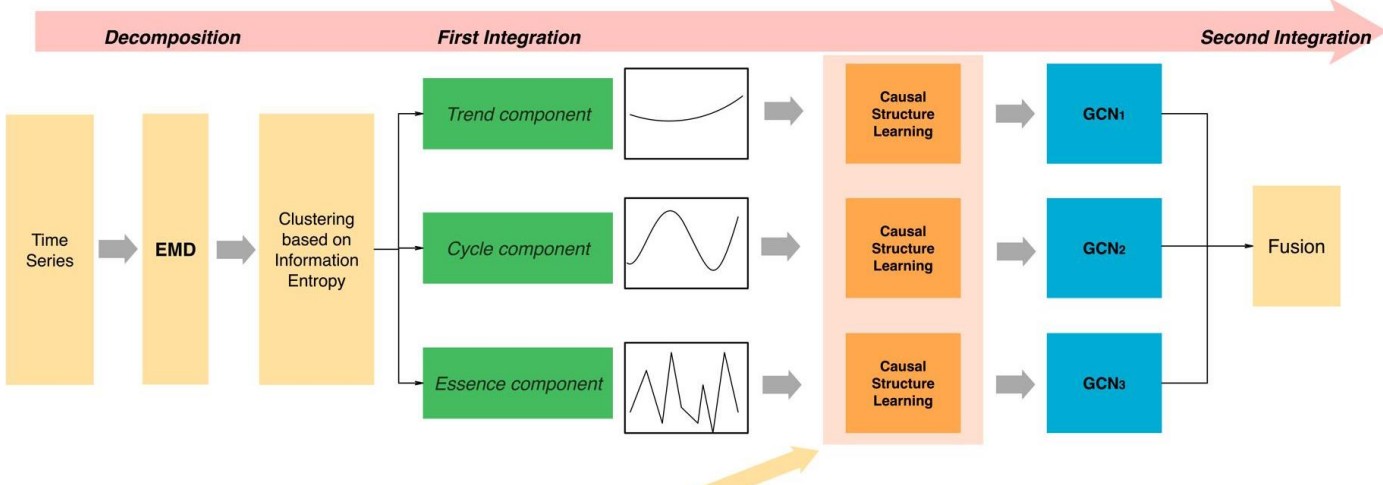

**Figure 2.** Structure of DIC-ST.

## 2. Related Works

Many cities are facing the challenges brought by rapid urbanization [12]. In recent years, research on smart cities has attracted extensive attention. J. Liu et al. mentioned that urban flow analysis is an important research content of smart city construction, in which urban flow model analysis focuses on the continuous state of urban flow. How to mine, store and reuse traffic patterns from urban multi-source heterogeneous big data is a challenge. Therefore, they proposed a regional flow pattern knowledge mining network to mine and store urban flow patterns. X. Pan et al. [13] indicated that wireless traffic can be used for smart city research. They pointed out that smart cities need to make full use of information technology to respond intelligently to all needs, including networks and urban services. They proposed a differential evolution back propagation (DE-BP) neural network traffic prediction model suitable for intelligent city network to predict network traffic. Similarly, S. Jiang et al. [14] transformed ubiquitous mobile phone data into interpretable human mobility patterns, taking Singapore as an example. Through the development of data mining pipeline, the spatial distribution of residents' travel patterns in different urban areas was quantified. The ultimate goal was to help planners effectively acquire urban knowledge from big data to target specific urban areas to improve future infrastructure and service planning. V. Javidroozi [15] pointed out that in order to develop a smart city, it is necessary to integrate all components of the city into a system. As a technology, urban computing can solve the complexity of providing appropriate services to citizens through different urban departments or systems, which promotes this. Similar to some other urban computing studies [13,14,16], we also use wireless network data in urban computing research.

In the process of data processing, data mining based on a decomposition and integration system is proposed. EMD is used to decompose the original data. EMD is a technique

of signal feature extraction, which can decompose time series data into finer-grained components [17]. As a timing processing technology, EMD has been widely used. In [18], aiming at the research of meteorological index prediction, the authors put forward three prediction models: a hybrid prediction model based on residual prediction, a hybrid prediction model based on EMD and a hybrid prediction model based on EMD and residual prediction. The authors tested three models. The experimental results showed that prediction accuracy of three models was significantly improved compared with the traditional model. The third model is a hybrid prediction method based on EMD and residual prediction, and its prediction performance was the best. The reason was that EMD makes the original sequence stable, and prediction performance of this model on stable data is generally good. R. K. Jalli et al. [19] proposed a hybrid short-term wind speed prediction method based on EMD and random vector function chain network. Firstly, the chaotic historical wind speed data is decomposed into several IMFs by EMD. These IMFs are used in the proposed prediction model. Finally, the effectiveness was confirmed. In [20], motivation of the research is that IMF allows to calculate the meaningful Hilbert transform of non-stationary data, from which the instantaneous time-frequency representation can be derived. Our spatiotemporal intrinsic mode decomposition method uses spatial correlation to extend the extraction of IMF from one-dimensional signals to multi-dimensional signals. It can be seen that most studies put the subsequences decomposed of EMD into different models. In this paper, after EMD decomposition, we integrated and extracted three components in the timing. This step ensured that the characteristics of each subsequence are consistent, so that the same prediction model can be used. At the same time, the complexity was greatly reduced.

Three components extracted by decomposition and integration are used for causal structure learning, respectively. Causal data mining has also been a research hotspot in recent years. In the research of data association, the focus of research has gradually shifted from correlation analysis to causality verification. In the real world, multivariate time series (MTS) data are common in various fields. Existing methods assume that the value to be predicted of a single variable is related to all other variables in MTS data. In [21], they proposed a solution by using causality information as a priori knowledge. Moreover, they proposed the framework that considers multivariate time series as a graph structure with causality. A Papana et al. [22] mentioned that concept of Granger causality is increasingly applied to the characterization of directional interaction in different applications. In order to explain all the available information in multivariable time series, a multivariable framework for estimating Granger causality is very important. In [23], Granger causality analysis based on neural network was used for root cause diagnosis, which effectively solved the problem that Granger causality analysis based on linear model can not process nonlinear data. In [24], a human behavior event analysis and calculation model integrating perceived causality was proposed. W. Tian designed and established a causal rule representation method based on default logic. Research in [25] aimed to determine the causes of air pollutants in surrounding cities that affect air quality. They proposed a compressed sensing causality analysis method by combining Granger causality analysis and maximum correlation entropy criterion to effectively identify the causality of air pollutants between Beijing and surrounding cities.

Studies on time series has lasted for a long time, and many methods have been used in urban computing, especially in traffic prediction. However, existing research has not deeply explored the time series, especially combined with causality test. In this paper, three components of cellular traffic data were extracted, respectively, causal structure learning for three components was applied, and the application scenario of each component was given. Finally, a prediction system based on decomposition integration and causal structure learning was proposed.

## 3. Materials and Methods

### 3.1. Data Description

With the development of technology and the expansion of telecom market, mobile data detection system can collect network signaling and user's service information extraction and processing data mining and analysis provide operators with a very valuable data model. In the current 5G commercialization era, facing the challenge of massive data, mobile data detection system needs to be able to collect data properly to meet the service response requirements and the application requirements of smart city. Data acquisition has always been a means for operators to effectively optimize the network. Through optimization, key indicators that directly affect users' subjective experience such as connection rate, call drop rate and voice quality can be improved, so as to provide users with reliable, stable and high-quality network services for quality of life changes in smart city.

When mobile phone and the base station conduct service, multiple types of mobile communication data with positioning information will be generated. These data include both event records and service records. In cellular networks, the event types of mobile communication data mainly include receiving calls, sending and receiving short messages, connection of data links and so on. This information will be recorded with the user's use and every communication between the mobile phone and the base station, and the data will be directly accessed to the server by base station. These data are stored in the base station and are usually summarized and counted every hour. In this paper, 100 days of data are used for method validation. The data structure of cellular network was shown in Table 1. Real wireless network traffic was collected at 22 base stations in an urban area. The traffic data analyzed in this paper is downlink traffic of the base station. According to the longitude and latitude information of base stations, combined with Voronoi [26] algorithm, we can obtain the approximate coverage of the base station. Among them, the Voronoi algorithm was widely used in base station coverage area estimation [27–29]. On this basis, we applied wireless network data to the research of urban computing. The coverage of 22 base stations is shown in Figure 3. Urban population analysis based on mobile communication data is essentially a data mining process, which usually includes data collection, data preprocessing, data mining and data visualization. Proposed method for data mining in the following chapters will be introduced in detail.

**Table 1.** Comparison of prediction methods.

| Timestamp | eNodeB ID | Average Number of Users Conected to the eNodeB | Maximum Number of Users Conected to the eNodeB | Uplink Traffic (GB) | Downlink Traffic (GB) |
|---|---|---|---|---|---|
| *2019/7/1 0:00* | *1* | 77.872 | 145 | 0.8802 | 5.5389 |
| *2019/7/1 10:00* | *1* | 67.8724 | 124 | 0.7015 | 4.0745 |
| | | ... | | | |
| *2019/7/1 22:00* | *1* | 57.1325 | 115 | 0.5501 | 2.1941 |
| *2019/7/1 23:00* | *1* | 59.9522 | 130 | 0.1145 | 2.1351 |
| | | ... | | | |
| *2019/7/1 0:00* | *22* | 133.8174 | 973 | 1.3402 | 11.7154 |
| *2019/7/1 10:00* | *22* | 97.4827 | 780 | 1.0155 | 7.2151 |
| | | ... | | | |
| *2019/7/1 22:00* | *22* | 139.0239 | 859 | 0.9138 | 12.1186 |
| *2019/7/1 23:00* | *22* | 101.9206 | 619 | 0.6791 | 8.1094 |
| | | ... | | | |

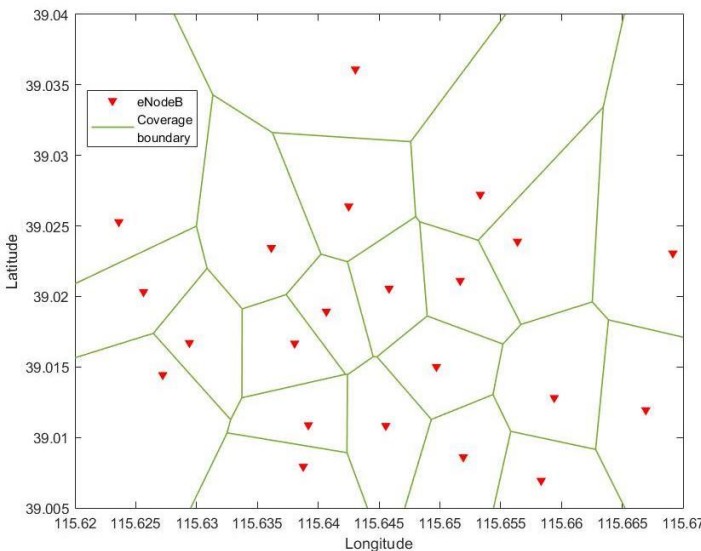

**Figure 3.** Coverage of base stations in the area.

*3.2. Methods*

3.2.1. Decomposition Integration System

In complex systems, time series often show the characteristics of multi-dimensional data. These complex features make time series prediction difficult. A single model can no longer meet the requirements of complex system analysis and prediction. In recent years, hybrid models have attracted more attention. In this paper, we used different time series analysis for three different components, and finally integrated the results. In decomposition and integration system, multi-scale decomposition was used to process time series data, so as to reduce the modeling difficulty of complex system and improve the analysis and prediction performance of the model. Specifically, the decomposition integration method decomposed the complex system into subsystems that are easy to analyze, which can significantly reduce the difficulty of prediction and improve the analysis and prediction performance of the model. The decomposition integration system in this paper included three steps: empirical mode decomposition, preliminary integration based on K-nearest neighbor (KNN) and prediction result integration.

Hilbert–Huang transform (HHT) is an empirical data analysis method. This method is adaptive. HHT consists of two parts, empirical mode decomposition (EMD) and Hilbert spectrum analysis (HSA) [30]. This technique is feasible for analyzing nonlinear and non-stationary data [31]. To fully describe EMD, we first define the IMF. IMF in EMD algorithm should meet two conditions [32]:

1. The number of extreme points (including local maximum points and local minimum points) is equal to the number of zero crossings or the difference is 1.
2. At any point, the average value of the envelope of local maximum and local minimum is 0. Different from the modal components in other decomposition methods, IMF is a generalized harmonic function rather than a simple fixed function, and its amplitude and frequency change with time.

EMD algorithm assumes that time series data generally contain multiple volatility components, and different volatility functions can be solved through the column part of the screening process and expressed by the IMF function [32]. EMD decomposition is based on the following assumptions: there is at least one maximum and minimum in the decomposed data. The local characteristic time scale is determined by the time interval between levels. If the data has no extreme points but contains inflection points, the extreme values can be obtained by one-order or multi-order differentiation.

Traffic data set can be expressed as $S = [S_1, S_2, ...S_n...S_N]$, where $N$ is the number of base stations in the region. In this paper, $N = 22$. For any $S_n$ contained in $S$, we perform

EMD empirical mode decomposition. The HHT based EMD process for traffic data is as follows:

1. Identify all maximum and minimum values existing in $S_n$, which form upper envelope $M_U$ and lower envelope $M_L$;

2. The average value of envelopes can be expressed as $m_1$, which is obtained by Formula (1):

$$m_1(t) = (M_U + M_L)/2 \tag{1}$$

3. The initial component $h_1$ can be obtained by Formula (2):

$$h_1(t) = S_n(t) - m_1(t) \tag{2}$$

4. Calculate the envelope average value $m_{11}$ of $h_1$;
5. $h_{11}$ can be obtained according to $m_{11}$, as shown in the Formula (3):

$$h_{11}(t) = h_1(t) - m_{11}(t) \tag{3}$$

6. After repeating the extraction process for $K$ times, $h_{1k}$ becomes an IMF, which can be expressed as:

$$h_{1k}(t) = h_{1(k-1)}(t) - m_{1k}(t) \tag{4}$$

which can be also expressed as:

$$c_1(t) = h_{1k}(t) \tag{5}$$

7. Separate $C_1$ from $S$, and the remaining data can be expressed as:

$$r_1(t) = S_n(t) - c_1(t) \tag{6}$$

8. By repeating the same operations from step 1 to step 6, a plurality of decomposed components $c_1(t), c_2(t)...r_1(t), r_2(t)...$ can be obtained.

### 3.2.2. Components Extraction by Clustering

We integrate multiple IMFs for the first time by clustering. We hope to extract three components from the original time series: trend component, periodic component and essential component. Trend component has the smallest entropy, and the trend component represents the overall growth or decline trend of the data. The essential component is the main content of wireless network data. The purpose of communication is to transmit the uncertainty of information. Therefore, wireless network data, especially traffic data, has strong randomness, so the entropy of essential component is the highest. The periodic component depends on the tidal characteristic of cellular traffic and has a certain law. The information entropy of the periodic component is between the essential component and the trend component. For any IMF $c_n$, the information entropy is obtained by function 7. Where $P(c_n(t))$ is probability of $c_n(t)$.

$$H(c_n) = -\sum_t P(c_n(t)) log_2(P(c_n(t))) \tag{7}$$

At the same time, we notice that the frequencies of different components are also different. The trend component is low-frequency, the essential component has the highest frequency due to high randomness, and the periodic component is also between the two. Based on the principle of filter, we choose the number of peak points in the spectrum as the quantization index of frequency. For each IMF, the number of spectral peaks was obtained. We can first calculate the corresponding modulus through Fourier transform as shown in Functions (8) and (9), and then count the number of extreme points.

$$F_n(\omega) = \int c_n(t)^{-i\omega t} dt \tag{8}$$

$$F_n(\omega) = |F_n(\omega)| \tag{9}$$

In order to realize the adaptive ability of clustering algorithm for three components of different base station data and ensure the accuracy of clustering, KNN method was selected to classify the scatter points (information entropy, peak number of spectrum), as shown in Figure 4.

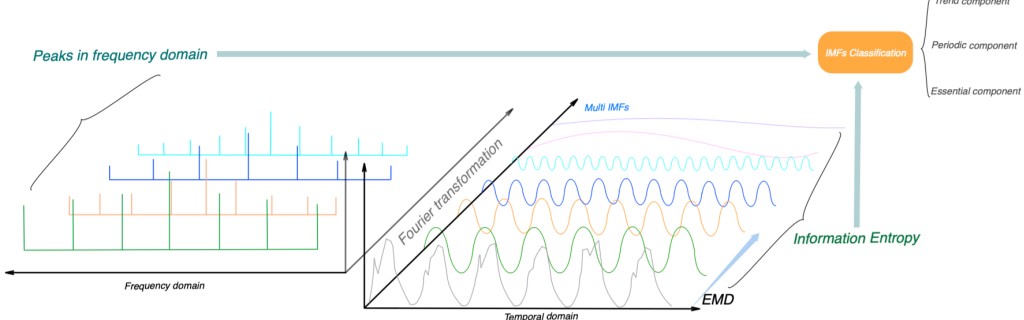

**Figure 4.** Data processing for KNN based clustering.

The coordinate points based on the number of peaks and information entropy in KNN are expressed as: $(I, P)$, where $I$ represents the information entropy and $P$ represents the number of peaks in the spectrum. The distance between two points was defined as Formula (10). Algorithm 1 presented components clustering process based on KNN. In Algorithm 1, the value of $K$ was set to 3, and three components are extracted: trend component $C_T$, periodic component $C_P$ and essential component $C_E$.

$$\rho = \sqrt{\left(I_i - I_j\right)^2 + \left(P_i - P_j\right)^2} \tag{10}$$

---

**Algorithm 1** Components Clustering.

---

1: **INPUT:** $c_i$ and $K = 3$
2: **Stage 1 Coordinate generation**
3: **for** $i = 1 : n$ **do**
4:　　Information entropy calculation for $c_i$ according to **Function (7)** and save the result in metrix $I$.
5:　　Count peak number $P_i$ according to **Functiona (8)** and **(9)** and save the result in metrix $P$.
6: **end for**
7: **Stage 2 Clustering based on KNN**
8: **for** $i = 1 : n$ **do**
9:　　**for** $j = 1 : n$ **do**
10:　　　$\forall I_i \in I, I_j \in I, P_i \in P, P_j \in P$.
11:　　　Distance calculation between $(I_i, P_i)$ and $(I_j, P_j)$ according to **Function (10)**.
12:　　**end for**
13: **end for**
14: Sort the distance in ascending order.
15: Select $K$ points with the smallest distance from the current point.
16: Determine the occurrence frequency of the category where the first $K$ points are located.
17: Return the category with the highest frequency of the first $K$ points as the classification result of current point.
18: Merge time series of the same class.
19: **OUTPUT:** $C_T$, $C_P$ and $C_E$.

---

### 3.2.3. Causal Structure Learning

Causal learning is a process of finding causal relationships in data structures. Reasoning about causal relationship between different time variables is easier than causal reasoning without a time structure [33]. In the causality mining part, we obtained the causal structure of three different components of the traffic data of each base station in the region. At present, the commonly used causality test algorithms mainly include Granger causality and transfer entropy. Granger causality is usually used in the context of linear structural equation. Transfer entropy is a nonlinear extension of Granger causality [34,35].

Due to the complexity of urban wireless network data, causal learning method based on transfer entropy was selected to process three components of each base station. Formula (11) gave the calculation method of transfer entropy, where $l$ is lag.

$$T_{C_i \to C_j} = H\left(C_i(t) \mid C_{i_{t-1:t-l}}\right) - H\left(C_i(t) \mid C_{i_{t-1:t-l}}, C_{j_{t-1:t-l}}\right) \tag{11}$$

In which,

$$H(C_i(t)) = -\sum_t P(C_i(t) log_2 P(C_i(t))). \tag{12}$$

The definition of causality judgment standard was given by Formula (13), where true represents that time series $C_i$ will affect the change of time series $C_j$. False means that the historical data of $C_i$ has no impact on time series $C_j$.

$$\frac{T_{C_i \to C_j}}{T_{C_j \to C_i}} = \begin{cases} true & \geq 1 \\ false & < 1 \end{cases} \tag{13}$$

After the extraction in the previous step, three components of the traffic of each base station were obtained. Causal structure learning was used to obtain the spatial correlation of each component. As shown in Algorithm 2, spatial causal structure of three components can be found. For three components, their causality graph can be expressed as: $D_T$, $D_P$ and $D_E$. The adjacency matrix based on causality graph can be expressed as: $A_T$, $A_P$ and $A_E$.

### 3.2.4. Prediction Models

Multivariate time series analysis considers multiple time series at the same time. Multivariate time series research is much more complex than univariate time series research, but in real life, decision-making or prediction often involves multiple related factors or variables. It is valuable to understand the relationship between these factors and give accurate prediction results of these variables. The purpose of multivariate time series analysis is to study the dynamic relationship between variables and improve the accuracy of prediction [36]. In this paper, for the prediction of different components obtained by the clustering part, we obtain the relationship between data of each base station by causal structure learning. Next, we make multivariate time series prediction based on causal structure learning for three components.

Graph convolution network (GCN) was used for time series prediction [37–39]. Unlike CNN, the input of GCN is graph structured data. CNN can obtain local spatial features, GCN which can process arbitrary graph structure data has attracted extensive attention [40]. In the application of GCN, graph embedding was applied to the prediction model. In recent years, embedding technology has attracted great attention in the field of machine learning and deep learning. Among them, the purpose of graph embedding is to use low dimensional and real value vectors to represent the nodes in graph and the relationship between nodes. Graph shows a two-dimensional relationship, while time series is one-dimensional data. Therefore, it is necessary to convert the graph into an embedded graph. Obtaining complex spatial correlation is a key problem in time series prediction. Most studies on spatial dependence in GCN for time series prediction are based on geographical distribution [40–42].

---

**Algorithm 2** Causal Structure Learning.

---

 1: **Stage 1 Causality Construction**
 2: **INPUT:** $C = [C_1, C_2...C_N]$ and N
 3: $\forall C_i \in C, C_j \in C$
 4: **for** $i = 1 : N$ **do**
 5:     **for** $j = 1 : N$ **do**
 6:         Trasfer entropy calculation between node $C_i$ and node $C_j$ according to **Function (11)**.
 7:         **if** $T_{C_i \to C_j} \geq 1$ **then**
 8:             Create a directed edge from node $i$ to node $j$.
 9:         **end if**
10:     **end for**
11: **end for**
12: **Stage 2 Causality Optimization**
13: **for** $i = 1 : n$ **do**
14:     **for** $j = 1 : n$ **do**
15:         **for** $k = 1 : n$ **do**
16:             **if** $T_{C_i \to C_k} \geq 1 \&\& T_{C_j \to C_k} \geq 1 \&\& T_{C_i \to C_j} \geq T_{C_i \to C_k}$ **then**
17:                 Delete the directed edge between node $i$ and node $k$.
18:             **else**Delete the directed edge between node $i$ and node $j$.
19:             **end if**
20:         **end for**
21:     **end for**
22: **end for**
23: **OUTPUT:** Causality Graph $D$.

---

In this paper, causal graph obtained by causal structure learning can not only describe the geographical attributes of data, but also imply the causal relationship between data. Structure of GCN in this paper is shown in Figure 5. When predicting each component of traffic of each area, we embed the structure of causality into the model. Causality graph can be expressed in the form of matrix. We assumed that each node in GCN represents the component time series of a base station, such as $x_{22}^{trend}$ in Expression (14), where 22 represents time series of trend component of the 22nd base station. A $T * 22$ matrix was used as input of GCN, where $T$ is the length of time series. Matrices of three components were shown in Expressions (14)–(16).

$$X^{trend} = \left[x_1^{trend}, x_2^{trend}......x_{22}^{trend}\right]^T \tag{14}$$

$$X^{periodic} = \left[x_1^{periodic}, x_2^{periodic}......x_{22}^{periodic}\right]^T \tag{15}$$

$$X^{essential} = \left[x_1^{essential}, x_2^{essential}......x_{22}^{essential}\right]^T \tag{16}$$

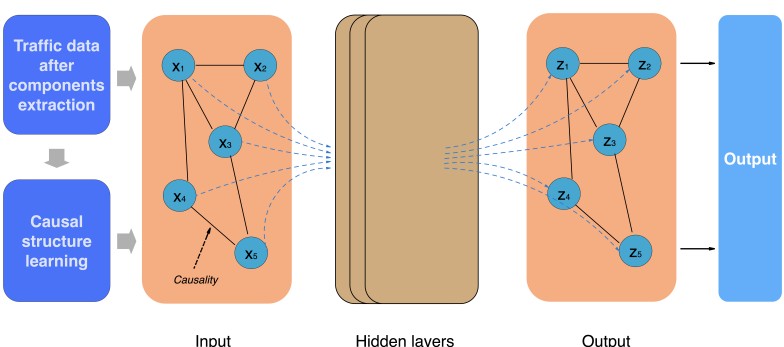

**Figure 5.** GCN structure.

A causal graph can be expressed in a form of matrix. $A$ is the matrix symbolizing causal structure. The spatial–causal-based adjacency matrix for any component can be expressed as Expression (17). Where $A_{1,22}$ represents the influence of the component of traffic for the 1st base station on the component of traffic for the 22nd base station. If there is a causal relationship between the two, that is $\frac{T_{C_1 \to C_{22}}}{T_{C_{22} \to C_1}} \geqslant 1$, value of $A_{1,22}$ in the matrix was 1, otherwise it was 0.

$$A = \begin{bmatrix} A_{1,1} & ... & A_{1,22} \\ ... & ... & ... \\ A_{22,1} & ... & A_{22,22} \end{bmatrix} \tag{17}$$

$$Z = f(X, A) = \sigma\left(D^{-1/2} A D^{-1/2} H^{(l)} W^{(l)}\right) \tag{18}$$

$$D = \sum_j A_{ij} \tag{19}$$

Formula (18) described expression of GCN. Where $X$ is input data, $D$ is degree matrix of $A$, $H$ is the feature of each layer, $W$ is weight matrix, $l$ is number of GCN layers, and $\sigma$ is the activation function. $Z$ is the final output of the network. In our verification, $l$ was set to 2. Prediction results of three components can be obtained by GCN. In model training, least square error was used as loss function and it can be given by Expression (20). Finally, these results were combined into the final traffic forecast value, which was given by Equation (21). Where $Z_{trend}$, $Z_{periodic}$ and $Z_{essential}$ are the prediction results obtained by GCN of three components, respectively.

$$L = \sum_{i=1}^{n} (Y_{true} - Y_{prediction})^2 \tag{20}$$

$$Y_{prediction} = Z_{trend} + Z_{periodic} + Z_{essential} \tag{21}$$

## 4. Results

### 4.1. Evaluation for Components Clustering

As described in Section 3.2.2, based on the time series processing framework of decomposition and integration, a time series extraction method based on KNN was proposed. Take traffic data collected by a base station in the area as an example, as shown in Figure 6. After EMD decomposition, traffic data was decomposed into 10 IMFs as shown in Figure 7.

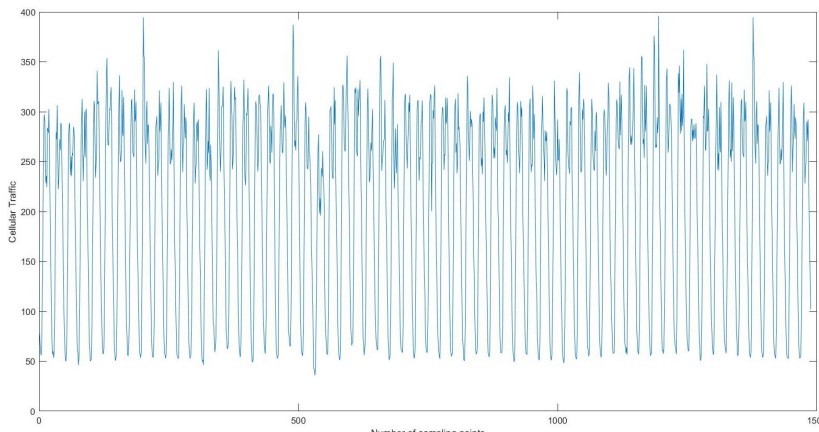

**Figure 6.** Sixty-two days of traffic of one base station.

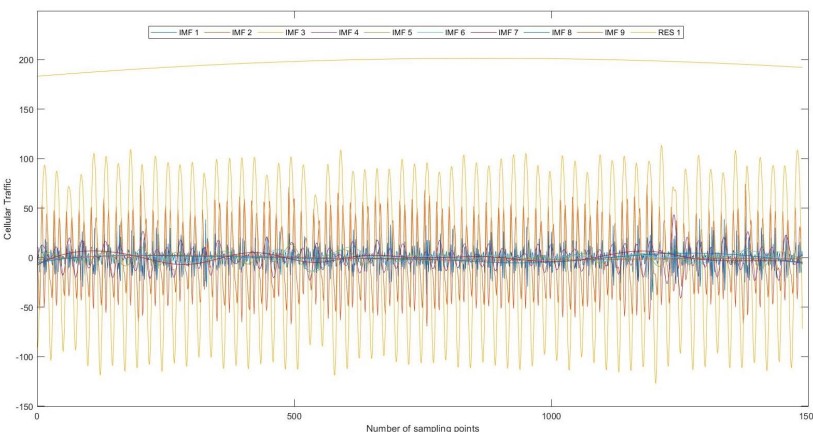

**Figure 7.** Multiple IMFs through EMD.

For traffic data of each base station in the area, multiple IMFs can be obtained after EMD decomposition. Information entropy and the number of peaks in spectrum of these IMFs were calculated, so that the scatter diagram based on entropy and spectrum can be obtained. KNN based classification was used for these scatter diagrams, and final clustering results are shown in Figure 8. Figure 8 shows the IMFs clustering results after the traffic decomposition of the four base stations in the area. According to such clustering results, IMFs of same category are integrated together to form a new component. Three components in Figure 9 showed the final result of components extraction of traffic in Figure 6.

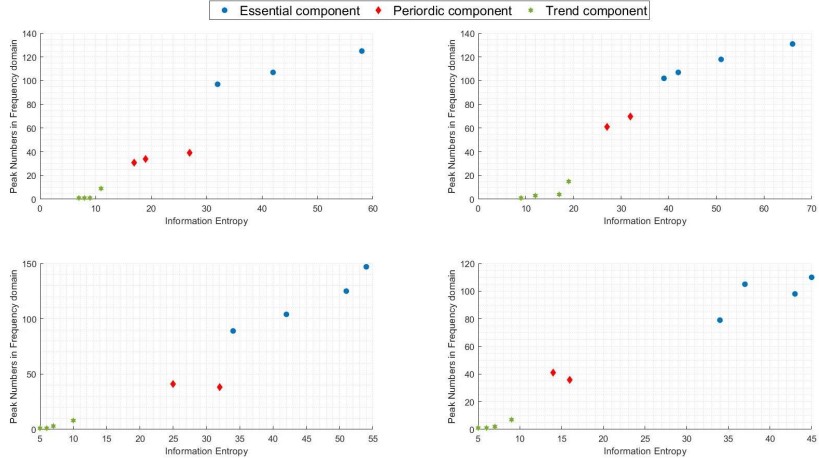

**Figure 8.** Clustering based on information entropy and frequency.

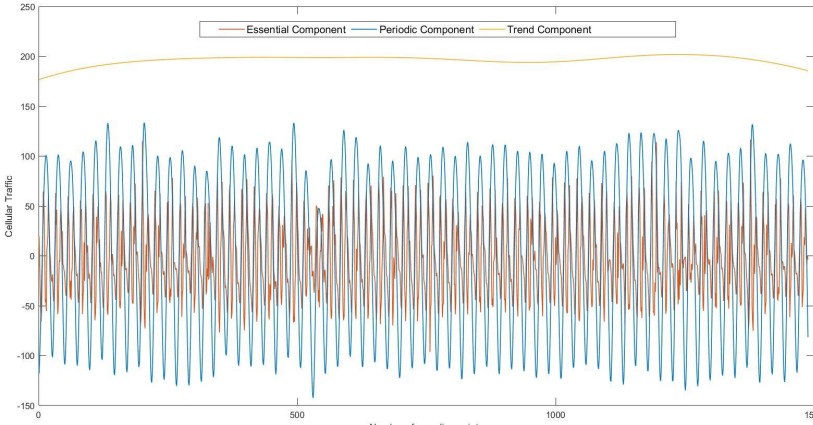

**Figure 9.** Three components after clustering.

As can be seen from Figure 9, trend component of the traffic was relatively stable, and there was no obvious increase or decrease trend for this base station. Periodic component had typical tidal characteristics. Randomness of essential component was strong, which is one of the key factors affecting the quality of network.

### 4.2. Evaluation for Causal Structure Learning

Figure 10 showed causality relationship between regions for three components. Causality of trend component combined with remote sensing map can provide guidance for urban planning. We found that this urban area is developing to the southeast. For the periodic component, the direction of causality was spread from center of the city to the periphery. The causality of periodic component can be used for crowd flow prediction and traffic management. The causality of essential component has no regularity. Wireless network is the most frequently used in people's daily life. Three components can be comprehensively considered in network optimization and management. Deep causal data mining provided data support for urban computing. At the same time, this causality will also be used for subsequent prediction in order to improve the accuracy of prediction.

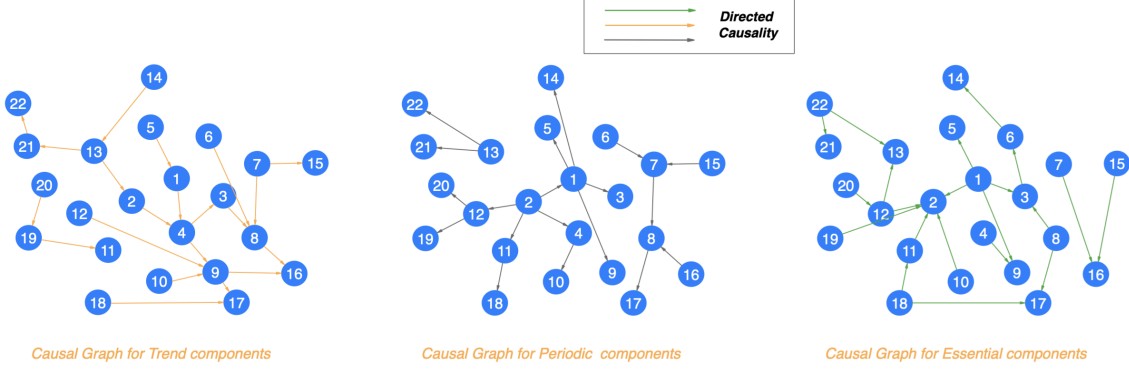

**Figure 10.** Causal graph for three components.

### 4.3. Prediction Performance

Through causal structure learning, causal structure graph of three components were obtained. In order to apply this causality to prediction, we transformed prediction into a graph to graph problem. The directed acyclic graph learned through causal structure can be described by matrix, which is causal matrix. We put causal matrix into GCN as adjacency matrix. When predicting the traffic through GCN, the ratio of training set to test set was 3:1. The data of 10 days was used to predict. The batch size in GCN was set to 128 and the number of iterations was 300. After obtaining prediction results of three components, we finally integrated these results.

In order to verify impact of causal structure learning on prediction performance, BIC-ST(no causality), which means prediction method based on decomposition integration and GCN, was also used to predict traffic. ARIMA, as an algorithm often used for traffic prediction [40–42] was also used for algorithm comparison. The prediction results of three base stations in the area were used as examples as shown in Figures 11–13.

Furthermore, in order to more clearly describe performance of different algorithms, RMSE and MAPE were calculated according to Formulas (21) and (22) for the results obtained by different methods. The verification results are given in Table 2. Through these results, it can be found that causal structure learning plays a role in improving prediction performance. At the same time, proposed method can be well used for network traffic prediction. In addition to this, Figure 14 showed training loss function versus training iterations. It can be seen from the figure that when the number of iterations is 150, the curve tends to be stable. When the number of iterations is close to 300, the loss function no longer fluctuates.

$$RMSE = \sqrt{\frac{1}{n} \sum_{i=1}^{n} \left( Y_i' - Y_i \right)^2} \tag{22}$$

$$MAPE = \frac{100\%}{n} \sum_{i=1}^{n} \left| \left( Y_i' - Y_i \right) / Y_i \right| \tag{23}$$

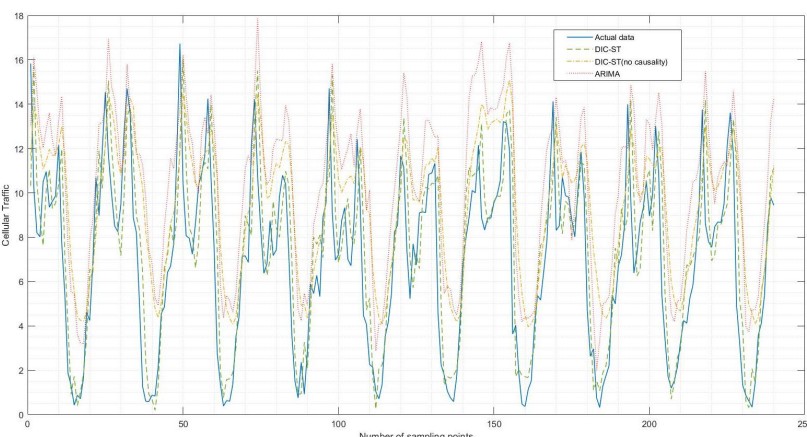

**Figure 11.** Prediction performance comparison for base station 1.

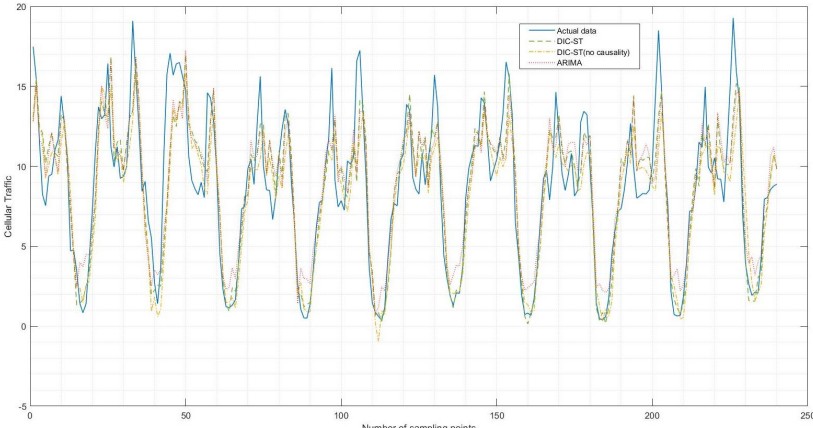

**Figure 12.** Prediction performance comparison for base station 2.

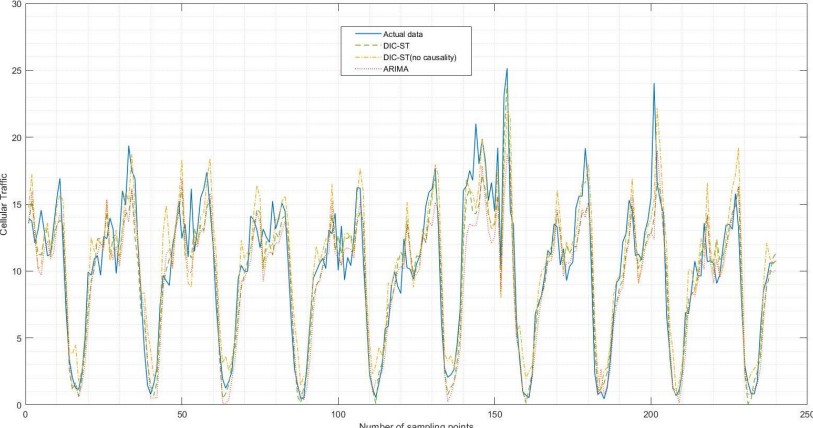

**Figure 13.** Prediction performance comparison for base station 3.

**Table 2.** Comparison of prediction methods.

| Methods | Base Station 1 | | Base Station 2 | | Base Station 3 | |
|---|---|---|---|---|---|---|
| | RMSE | MAPE | RMSE | MAPE | RMSE | MAPE |
| DIC-ST | 2.1952 | 43.5132 | 2.141 | 23.7567 | 1.9206 | 21.2691 |
| DIC-ST (no causality) | 3.5753 | 97.5417 | 2.3824 | 29.2335 | 2.4058 | 40.009 |
| ARIMA | 4.1138 | 104.2443 | 2.3028 | 46.5367 | 2.3538 | 25.2774 |

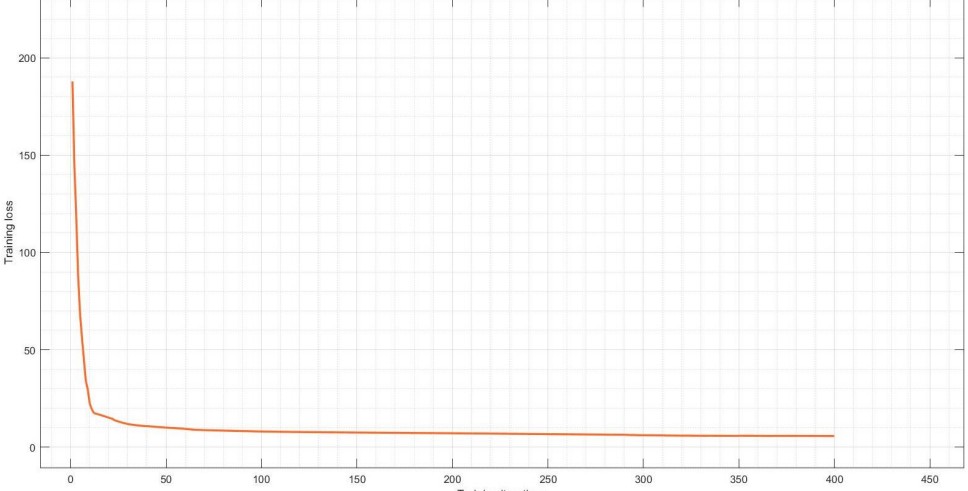

**Figure 14.** Training loss versus training iterations.

## 5. Discussion

KNN based on entropy and spectrum was used to cluster multiple IMFs obtained by EMD. As can be seen from Figure 8, joint analysis of information entropy and spectrum information ensured the accuracy of clustering. This time series extraction method converted original traffic data into three components. Among them, trend component can be used in urban planning and network construction. By combining with remote sensing information, long-term mobility of people can be clearly obtained. The combination of periodic component and remote sensing map is important for intelligent transportation management. Compared with public transportation data, periodic component of wireless network data can more clearly describe the short-term mobility of people. Research on essential component can provide data support for wireless network management, so as to provide users with better quality of service.

In this paper, causal structure learning was combined with time series decomposition. This is the first deep causal mining of traffic data. Figure 10 showed different causal structures of three components. Causal structure of each component is different. For different research fields in urban computing, this deep causal mining improved the availability of data. This method has greater value than traditional data analysis methods.

Finally, graph neural network was selected and used for three components, respectively, according to their respective spatial causal structure. The function of graph neural network is to embed the spatial causal structure into prediction model in the form of graph. Combined with actual network data, results showed that causal relationship between regions can significantly improve the prediction performance. Proposed DIC-ST was valuable in wireless network management and urban management.

## 6. Conclusions

Urban computing based on wireless network data is an important means of building smart city, which can provide solutions for various applications of smart city. Data of cellular network has randomness and structural difference. In this paper, traffic of cellular

network was deeply mined based on causal structure learning, and DIC-ST, a method for network traffic prediction, was proposed. After components extraction, three components of cellular traffic can be used in different fields of urban computing. When these three components were combined with causal structure learning, the prediction performance was significantly improved. Combining base station coverage estimation with the remote sensing map, the accurate prediction of traffic data can be applied to many aspects of urban computing. The method proposed in this paper can be well applied to urban planning, traffic management and network optimization. This is helpful for smart city construction and improving the quality of life of urban residents.

Our future work will expand the scope of data acquisition. Big data analysis is mainly reflected in the wide range of data sources and large amount of data. We hope to expand data sources, such as weather, social networking information, etc. Urban computing based on data fusion will be the focus of our research.

**Author Contributions:** Conceptualization, K.Z.; methodology, K.Z.; software, S.M.; validation, J.Z. and Z.S.; formal analysis, K.Z. and X.C.; resources, G.C.; data curation, K.Z.; writing—original draft preparation, K.Z.; writing—review and editing, G.C. and S.M.; visualization, X.C.; supervision, G.C. All authors have read and agreed to the published version of the manuscript.

**Funding:** This work was supported by the National Key Research and Development Project of China under Grant 2020YFB1806703.

**Institutional Review Board Statement:** Not applicable.

**Informed Consent Statement:** Not applicable.

**Data Availability Statement:** Not applicable.

**Conflicts of Interest:** The authors declare no conflict of interest.

## Abbreviations

The following abbreviations are used in this manuscript:

| | |
|---|---|
| MTS | Multi Time Series |
| DIC-ST | Spatial Temporal time series prediction based on Decomposition and Integration system with Causal structure learning |
| EMD | Empirical Mode Decomposition |
| HHT | Hilbert–Huang transform |
| IMF | Intrinsic Mode Function |
| KNN | K-Nearest Neighbor |
| GCN | Graph Convolution Network |
| ARIMA | Autoregressive Integrated Moving Average mode |
| CDR | Call Detail Record |

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
