# Peer review of "DIC-ST: A Hybrid Prediction Framework Based on Causal Structure Learning for Cellular Traffic and Its Application in Urban Computing"

_remotesensing, doi:10.3390/rs14061439_

Round 1
Reviewer 1 Report
The novelty of the proposed paper is highlighted by the proposed deep causal mining method for time series that integrate different components to sort out the traffic relationship of the analyzed regions to provide better predictions. The authors have proposed a novel time series analysis method DIC-ST and the results show that the proposed method improves the accuracy of cellular traffic prediction.
The proposed paper is significant to the field of urban computing as it involves various fields such as urban planning, network optimization, energy consumption. The proposed DIC-ST hybrid prediction framework based on causal structure learning for cellular traffic has an important role for urban computing and associated research areas.
The paper follows the standard format of a research paper and is well documented with related work references regarding urban computing, causal structure learning, smart city concept and other related aspects of the proposed paper. It has an extended introduction regarding urban computing and the importance of applying various technologies in the development of modern smart city and urban planning concepts. Within the introduction there are presented elements regarding wireless network data and time series predictions applied to crowd mobility, geographical zoning, urban planning, development, and security. The related works section presents multiple recent work studies related to various challenges associated with rapid urbanization and smart cities. The materials and methods section provides sufficient details regarding the data collection. The authors have monitored the traffic of 22 base stations within an urban area and the traffic data analyzed within the proposed paper is downlink traffic of the base station. The data mining section is presented within the following subsection, the authors have proposed an components extraction algorithm with three outputs. For the causal structure learning process, the authors have created an algorithm that has two stages (Causality Construction and Causality Optimization) and outputs the causality graph. The final subsection presents the proposed graph convolution network used to define the time series predication based on causal structure learning for the three components. The structure of the graph convolution network that can imply causal relationship between data is illustrated in Figure 5. The conclusions are based on the findings of the proposed method defined by the authors.
The proposed work has a high interest for readers and scientists that are working on urban computing which has implication in various research fields such as network management, intelligent transportation, base station siting, energy consumption that integrate remote sensing map information.
There is a type error in the article title – Urban Computing, not Unban Computing
Author Response
Dear Reviewer,
Thank you for your letter and for the reviewers’ comments concerning our manuscript entitled “DIC-ST: A Hybrid Prediction Framework based on Causal Structure Learning for Cellular Traffic and its Application in Urban Computing”. Thanks for your appreciation. Our future research will continue to focus on cellular network data based urban computing. A wider range of data sources will be introduced into future research. Your comments motivate us a lot. Thank you again for your review.
We corrected the type error in the article title.
If you have any question about this paper, please don’t hesitate to let me know.
Sincerely yours,
Kaisa Zhang, Gang Chuai, Jinxi Zhang, Xiangyu Chen, Zhiwei Si and Saidiwaerdi Maimaiti
kaisa@bupt.edu.cn

Reviewer 2 Report
The title is wrong -> I believe that the authors meant urban instead of unban
English is odd although not wrong.
Define urban computing -> the authors say how important urban computing is but there is no definition about what urban computing means.
By wireless network data the authors meant cellphone data overall?
Although the contribution is clear, the authors should clarify which problem that they are trying to solve in the context of urban computing - in the way that it is being show it is a deep learning problem using cellphone data - there is no problem with that. It is just not clear.
In the related works: "Although researches on time series has lasted for a long time, and many methods have been used in urban computing, especially in traffic prediction." -> this sentence is odd.
Regarding the proposal, it is the best part of the paper - but it is not clear the deployment of the algorithm - is it online? When it is evaluated? Since it is difficult to understand when the algorithm is going to be employed it is difficult to see how it is used.
The experiments lacked the platform on which they were developed. It was used python? What was the confidence interval? There are several details missing that prevent the experiments to be reproducible.
What were the features chosen?
Overall It seems as a good idea, but there is room for improvement before being ready for publication.
Author Response
Response Letter
(Manuscript ID: remotesensing-1598337)
Dear Reviewer,
Thank you for your letter and for the reviewers’ comments concerning our manuscript entitled “DIC-ST: A Hybrid Prediction Framework based on Causal Structure Learning for Cellular Traffic and its Application in Urban Computing”. These comments are all valuable and very helpful for revising and improving our paper, as well as the important guiding significance to our researches.
We thank the reviewer for the valuable remarks. They have helped us improve the presentation of the paper to avoid confusion and improve the overall quality. We respond to each of the review comments below.
We have carefully studied the review comments. We would like to express our sincere appreciation to the reviewers for providing the detailed review comments, which have helped us improve the technical content and the presentation quality of our manuscript.
We have considered each of the review comments very carefully, and revised our manuscript accordingly, and the revisions in the revised manuscript are marked in blue.
We hope that the revised manuscript, together with the detailed response to the review comments, will be found satisfactory by you. Please do not hesitate to contact me should you have any questions/concerns.
Sincerely yours,
Kaisa Zhang, Gang Chuai, Jinxi Zhang, Xiangyu Chen, Zhiwei Si and Saidiwaerdi Maimaiti
kaisa@bupt.edu.cn
Response to Review
- Comments: Define urban computing -> the authors say how important urban computing is but there is no definition about what urban computing means.
Response: Thanks for your valuable comments. We apologize for our unclear presentation. The definition of urban computing has been added to the article and marked in red:
Urban computing is a new cross field of urban planning, transportation, energy, environment, economy and sociology based on computer science. The task of urban computing is to first perceive and obtain all kinds of big data generated in the city, and then analyze, process and display big data by using efficient data management technology, advanced algorithms and novel visualization method, so as to solve many problems and challenges existing in the city, such as traffic congestion, poor network quality, backward planning and so on. Urban computing combines sensing technology, advanced data management and analysis models and novel visualization methods to create a solution to improve urban environment, human life quality and urban operation system[1]. Among them, the discovery of mobile model is one of the most challenging problems in urban computing. It can improve the resource supply and management of cities [2].
- Comments: By wireless network data the authors meant cellphone data overall?
Response: We apologize for the unclear description. The data in this paper came from a Chinese operator. These data were collected from multiple base stations in the city. The information in data records the information of users (mobile phones) served by each base station. Collected data contain statistical value of traffic of all users in the coverage area of each base station.
- Comments: Although the contribution is clear, the authors should clarify which problem that they are trying to solve in the context of urban computing - in the way that it is being show it is a deep learning problem using cellphone data - there is no problem with that. It is just not clear.
Response: Thanks for your valuable comments. In this paper, we decomposed traffic of each region in the city into three components through proposed time series decomposition integration method, and then three components were used in causal structure learning respectively. The application of each component in urban calculation is slightly different.
This paper mainly focuses on three kinds of problems in urban Computing: mobile model, network quality and urban development. We decomposed time series into three components: periodic component, trend component and essence component. Each component plays a different role in urban computing. Research on trend component can be used in base station location, urban planning and so on; Periodic component symbolizes the law of population flow and can be used for the management of wireless network and monitoring of traffic information; The essential component is the key to ensure the quality of user service. We established a temporal-spatial model for each component and introduced causal learning into the model.
- Comments: In the related works: "Although researches on time series has lasted for a long time, and many methods have been used in urban computing, especially in traffic prediction." -> this sentence is odd.
Response: Thanks for your valuable comments. We apologize for our unclear expression. This sentence has been changed to ‘Researches on time series has lasted for a long time, and many methods have been used in urban computing, especially in traffic prediction.’ and marked in red.
- Comments: Regarding the proposal, it is the best part of the paper - but it is not clear the deployment of the algorithm - is it online? When it is evaluated?
Response: Thanks for your valuable comments. Our proposed method is offline learning. Collected historical data was used to predict future value, and predicted value can be used in many aspects of urban computing. When evaluating the performance of the method, we compared predicted value with the actual value. In the future, our research will consider online prediction scheme, and the online window function setting will be the focus of our next research.
- Comments: Since it is difficult to understand when the algorithm is going to be employed it is difficult to see how it is used.
Response:
Thank you for your careful review. This is a very meaningful question. In this paper, the purpose of our research is to extract three components of cellular traffic through proposed time series decomposition and integration method. Combined with causal structure learning, we find that each component had different significance for urban computing. Compared with the end-to-end prediction method, proposed algorithm maximized the value of data while ensuring the prediction performance.
- Comments: The experiments lacked the platform on which they were developed. It was used python? What was the confidence interval? There are several details missing that prevent the experiments to be reproducible.
Response: Thanks for your valuable comments. We used python in tensorflow environment for method evaluation. Our simulation parameters are set as follows:
The batch size in GCN was set to 128 and the number of iterations was 300. Least Square Error was used as loss function, and it can be expressed as function (1). We added this part to the article as follows:
(1)
- Comments: What were the features chosen?
Response: Thanks for your valuable comments. In this paper, features represent three components in time series. We combine time series feature extraction with causal structure learning to maximize the value of data analysis in the research of urban computing.
Reference:
- Zheng, "Urban Computing: Tackling Urban Challenges Using Big Data," 2016 IEEE 24th International Requirements Engineering Conference (RE), 2016, pp. 3-3, doi: 10.1109/RE.2016.14.
- Altomare, E. Cesario, C. Comito, F. Marozzo and D. Talia, "Trajectory Pattern Mining for Urban Computing in the Cloud," in IEEE Transactions on Parallel and Distributed Systems, vol. 28, no. 2, pp. 586-599, 1 Feb. 2017, doi: 10.1109/TPDS.2016.2565480.

Reviewer 3 Report
The paper objective is to predict future network traffic BW based on the historical data.
I have the following comments:
- It is not clear how figure 1 is useful to sort out mobility as stated
- Any reference for hybrid models mentioned in line 56.
- Equation 2, what is initial component?
- what are h11 and m11 in equation 3?
- Entropy proportionality to information generally is a fact, however it is not clear how this is related to this study. In section 3.2.2. The statement of "so the entropy of essential component is the highest" needs to be clarified.
- The rationale of equation 11
- What is the cost (loss) function of the deep network used
- Can the authors add plots that show the training loss function versus training iterations?
- Have the authors studied recurrent Neural Networks for this time series prediction instead? End-to-end solution may achieve similar results.
Author Response
Response Letter
(Manuscript ID: remotesensing-1598337)
Dear Reviewer,
Thank you for your letter and for the reviewers’ comments concerning our manuscript entitled “DIC-ST: A Hybrid Prediction Framework based on Causal Structure Learning for Cellular Traffic and its Application in Urban Computing”. These comments are all valuable and very helpful for revising and improving our paper, as well as the important guiding significance to our researches.
We thank the reviewer for the valuable remarks. They have helped us improve the presentation of the paper to avoid confusion and improve the overall quality. We respond to each of the review comments below.
We have carefully studied the review comments. We would like to express our sincere appreciation to the reviewers for providing the detailed review comments, which have helped us improve the technical content and the presentation quality of our manuscript.
We have considered each of the review comments very carefully, and revised our manuscript accordingly, and please refer to the PDF file for specific modification reply.
We hope that the revised manuscript, together with the detailed response to the review comments, will be found satisfactory by you. Please do not hesitate to contact me should you have any questions/concerns.
Sincerely yours,
Kaisa Zhang, Gang Chuai, Jinxi Zhang, Xiangyu Chen, Zhiwei Si and Saidiwaerdi Maimaiti
kaisa@bupt.edu.cn

Round 2
Reviewer 3 Report
The authors have addressed my concerns. I feel it's better to look into the following as well:
- Please add the entropy clarification mentioned in the response letter to the manuscript
- How the probability function calculated, that is used to calculate equation 11 and 12? Is it just based on numerical data?
- In Algorithm1, can the authors elaborate more on the difference between stage 1 and 2. Generally, it makes sense to have some points with known clusters and use KNN to cluster the unknown points according to the distance to the known points. However, it isn't clear how that is applied here. In other words, how the known points were obtained in the first place. Also, please comment on why clustering is needed
- What is the value of the time series length T?
- For training the GCN, is Y the future value of x? Please clarify that.
Author Response
Response Letter
(Manuscript ID: remotesensing-1598337)
Dear Reviewer,
Thank you for your letter and for the reviewers’ comments concerning our manuscript entitled “DIC-ST: A Hybrid Prediction Framework based on Causal Structure Learning for Cellular Traffic and its Application in Urban Computing”. These comments are all valuable and very helpful for revising and improving our paper, as well as the important guiding significance to our researches.
We thank the reviewer for the valuable remarks. They have helped us improve the presentation of the paper to avoid confusion and improve the overall quality. We respond to each of the review comments below.
We hope that the revised manuscript, together with the detailed response to the review comments, will be found satisfactory by you. Please do not hesitate to contact me should you have any questions/concerns.
Sincerely yours,
Kaisa Zhang, Gang Chuai, Jinxi Zhang, Xiangyu Chen, Zhiwei Si and Saidiwaerdi Maimaiti
kaisa@bupt.edu.cn
Response to Review
- Comments: Please add the entropy clarification mentioned in the response letter to the manuscript
Response: Thanks for your valuable comments. The clarification to entropy as mentioned in the response letter was placed in the first paragraph of 3.2.2.
- Comments: How the probability function calculated, that is used to calculate equation 11 and 12? Is it just based on numerical data?
Response: Thanks for your careful review and valuable comments. probability function was based on numerical data. We are sorry we didn't give a detailed explanation. The explanation of probability P has been added to the manuscript and marked in red.
- 3. Comments: In Algorithm1, can the authors elaborate more on the difference between stage 1 and 2. Generally, it makes sense to have some points with known clusters and use KNN to cluster the unknown points according to the distance to the known points. However, it isn't clear how that is applied here. In other words, how the known points were obtained in the first place. Also, please comment on why clustering is needed
Response: Thanks for your valuable comments. We apologize for our unclear expression.
In algorithm 1, the first stage includes the calculation of spectrum peak point and information entropy; The second part is the clustering algorithm. In clustering, Euclidean distance was used for distance calculation, as shown in Formula 1. Among them, the coordinates of each point were obtained by the first part.
In order to realize the adaptive ability of clustering algorithm for three components of different base station data and ensure the accuracy of clustering, KNN method was selected to classify the scatter points (Information entropy, Peak number of spectrum).
We chose this clustering method because we want to improve the stability of classification by considering entropy and frequency. Accurate component partition method can be obtained.
- 4. Comments: What is the value of the time series length T?
Response: Thanks for your valuable comments. We apologize for our unclear expression. In this paper, the data of 100 days were used for verification of proposed method. Since the data was counted every hour, for one base station, total number of data is 100 * 24 = 2400, and the ratio of training set to test set is 3:1. In the final test, we predict the data in the next 10 days. In model training, T=1584 and in model testing, T=816.
- 5. Comments: For training the GCN, is Y the future value of x? Please clarify that.
Response: Thanks for your valuable comments. I'm sorry that our description was not rigorous. In the formula, Y represents the final prediction result, while , and are predict value of three components respectively, which are output by graph convolution network. We have modified the corresponding content and marked it in red in the manuscript.